# Interplay of Modifiable and Non-Modifiable Risk Factors for Diabetes Mellitus in Saudi Adults

**DOI:** 10.3390/diagnostics15192451

**Published:** 2025-09-25

**Authors:** Mohammad A. Jareebi, Ibrahim M. Gosadi

**Affiliations:** Department of Family and Community Medicine, Faculty of Medicine, Jazan University, Jazan 45142, Saudi Arabia; mjareebi@jazanu.edu.sa

**Keywords:** diabetes mellitus, nutrition, genetic factors, modifiable risk factors, non-modifiable risk factors, Saudi Arabia

## Abstract

**Background/Objectives:** Diabetes Mellitus (DM) remains a critical public health issue in Saudi Arabia, shaped by complex interactions among genetic, lifestyle, and sociodemographic factors. This study explores interplay of modifiable and non-modifiable determinants of DM among Saudi adults. **Methods:** An analytical cross-sectional study was conducted among 3411 adults aged 18 years and above in the Jazan region, southwest of Saudi Arabia, in May–June 2024. Data was collected via a structured, pretested questionnaire assessing sociodemographic, dietary patterns, physical activity, smoking habits, and family history of DM. Bivariate analysis and logistic regression were used to identify associations with self-reported diabetes. **Results:** Out of 3411 participants (1735 males and 1676 females), 424 (12.4%) reported DM. Diabetics were older (48 vs. 32 years), more often male, married, had lower education, had larger families, had higher BMIs, and exhibited more tobacco use (*p* < 0.05), and a family history of diabetes was strongly associated with diagnosis of DM (*p* < 0.001). Diabetics were more likely to choose low-fat meats, avoid sugary foods, and select low-fat products (*p* < 0.05). In multivariate analysis, predictors were age (OR = 1.07, 95% CI: 1.06–1.09), male sex (OR = 1.65, 95% CI: 1.26–2.16), family history (OR = 7.68, 95% CI: 5.67–10.57), traditional housing (OR = 1.82, 95% CI: 1.11–3.05), and whole grain intake (OR = 0.67, 95% CI: 0.52–0.85). **Conclusions:** DM in Saudi Arabia is driven by both inherited and behavioral risks. These findings support the urgent need for integrated, culturally tailored prevention strategies that combine early screening for individuals with higher risk. Targeted actions such as relevant lifestyle interventions can help reduce disease burden and align with Saudi Vision 2030 health priorities.

## 1. Introduction

Diabetes mellitus (DM) is a global public health crisis, with approximately 589 million adults aged 20–79 years living with the condition in 2024—equivalent to 11.1% of the global adult population [1], and this figure is projected to increase to 853 million by 2050, representing a 46% rise [2]. Alarmingly, 43% of adults with diabetes remain un-diagnosed, highlighting significant gaps in detection and care worldwide [1]. The Middle East and North Africa (MENA) region continues to be severely affected [3], and within this region, Saudi Arabia ranks among the top ten countries globally, with diabetes prevalence reaching 23.1% of adults, equating to approximately 5.34 million people in 2024 [4].

The burden of DM is influenced by a complex mix of modifiable and non-modifiable risk factors. Among modifiable determinants, body mass index (BMI), dietary habits, smoking, and physical inactivity play prominent roles [5]. Overweight and obesity—affecting nearly 60% of Saudi adults—are strongly associated with insulin resistance and the development of type 2 DM (T2DM) [6]. A study of Saudi adults revealed that 59% do not meet recommended physical activity levels, and over 70% consume insufficient fruits and vegetables, increasing their risk of metabolic disorders [7]. Furthermore, poor dietary patterns characterized by high intake of saturated fats, re-fined carbohydrates, and sugary beverages contribute significantly to diabetes risk [8]. Smoking, another modifiable behavior, has been shown to increase T2DM risk by 30–40%, with heavy smokers at even higher risk due to nicotine-induced insulin resistance and central obesity [9]. These lifestyle factors are of particular concern in Saudi Arabia, where rapid economic development has been accompanied by shifts toward sedentary behavior and Westernized diets [10].

Conversely, non-modifiable determinants such as age, genetics, family history, and dyslipidemia also contribute significantly to diabetes susceptibility [11]. Individuals with a first-degree relative with DM are two to six times more likely to develop the disease [12]. Genetic studies have identified more than 400 gene loci associated with T2DM, including TCF7L2, FTO, and SLC30A8, which influence insulin secretion and glucose metabolism [13]. Furthermore, dyslipidemia—a common metabolic disorder characterized by elevated LDL-C, triglycerides, and reduced HDL-C—has both hereditary and lifestyle-driven components and is frequently observed in individuals with or at risk of diabetes [14].

The genetic architecture of the Saudi population, shaped by high rates of consanguinity (estimated at 25–60%), may contribute to a higher expression of deleterious alleles associated with metabolic diseases, including diabetes [15]. This unique genetic profile necessitates a more nuanced understanding of how hereditary factors interact with modifiable risks such as diet and lifestyle. Some gene–diet interactions suggest that specific genetic variants may enhance or blunt the impact of dietary factors on glycemic control, underscoring the need for personalized nutrition strategies [16].

Evidence increasingly supports the interplay between modifiable and non-modifiable determinants in influencing diabetes risk. Individuals with a high genetic risk profile may significantly reduce their risk by maintaining a healthy lifestyle, including optimal diet and weight control [17]. Conversely, unhealthy behaviors may amplify genetic risk, accelerating disease onset. This dynamic interplay underscores the importance of integrative risk assessments and interventions that consider both inherited and behavioral factors.

Despite the global proliferation of diabetes research, key knowledge gaps persist, particularly within the Saudi population. Much of the existing literature either focuses on lifestyle or genetic risk factors in isolation, limiting insights into their combined effects. Moreover, many international studies are not fully generalizable to Arab populations due to distinct cultural, dietary, and genetic differences. Additionally, findings across studies have been inconsistent: some report strong gene–environment interactions, while others find minimal modulation by lifestyle behaviors [18]. In Saudi Arabia, few large-scale studies have comprehensively evaluated the combined effect of genetics, nutrition, BMI, smoking, and family history in relation to diabetes risk.

Therefore, this study aims to investigate the interplay between nutrition and genetic factors within the broader context of modifiable and non-modifiable determinants of DM among Saudi adults. Unlike previous studies that have examined these variables in isolation, our approach integrates both behavioral and hereditary factors to capture a more comprehensive risk profile. To our knowledge, this is one of the first large-scale studies in Saudi Arabia to simultaneously assess lifestyle behaviors, dietary patterns, BMI, smoking status, and genetic predisposition—including family history—within a unified analytical framework. By examining how these determinants interact, rather than operate independently, the study offers novel insights into the complex etiology of diabetes in this high-risk population and lays the groundwork for precision prevention strategies tailored to Saudi adults.

## 2. Materials and Methods

### 2.1. Study Design

This analytical, cross-sectional study was conducted between May and June 2024 among Saudi adults from the general population. Participants were recruited from various regions and represented a broad range of sociodemographic backgrounds, including different age groups, occupations, and education levels. Most had access to healthcare services through Ministry of Health facilities, private providers, or university hospitals, either via insurance or self-payment.

### 2.2. Study Population

Eligible participants included Saudi adults aged 18 years or older who were able to provide informed consent. Individuals diagnosed with type 1 DM, those with incomplete responses to key variables (e.g., diabetes status, BMI, or family history), and pregnant women were excluded from the study to avoid confounding factors related to glucose metabolism. The sample size was calculated using the standard formula for cross-sectional studies, assuming a 50% prevalence (*p* = 0.5), a 95% confidence level (Z = 1.96), and a 2% margin of error (d = 0.02), resulting in a minimum required sample size of 2401 participants. To account for potential 25% non-response, the target sample size was increased to 3411, strengthening the study’s statistical power.

### 2.3. Data Collection

A pretested, structured questionnaire was administered to gather data on sociodemographic, clinical, lifestyle, and dietary variables, along with family history of chronic diseases. The questionnaire was developed based on the WHO STEPS instrument for non-communicable disease risk factors [19]; physical activity was assessed using the International Physical Activity Questionnaire (IPAQ), which has demonstrated acceptable reliability and validity internationally [20] and in Arabic-speaking populations, including Saudi Arabia [21], and using validated nutrition assessment tools [22]. It included questions covering sociodemographic factors such as age, gender, education, income, social status, residence, and housing type. Anthropometric measurements were obtained through self-reported height and weight, which were then used to calculate body mass index (BMI) and categorized according to World Health Organization criteria [23]. Lifestyle behaviors assessed included smoking habits (both cigarettes and shisha), physical activity frequency and duration, and average sleep duration. Nutritional habits captured information on the consumption of whole grain products, fruit and vegetable intake (≥5 servings per day), preference for low-fat meats and dairy products, and avoidance of foods high in sugar. In this context, low-fat meats referred to items such as skinless chicken and lean cuts of beef or lamb, while low-fat products referred to low-fat milk, yogurt, and cheese. Additionally, non-modifiable and genetic factors were examined through questions about family history of DM and dyslipidemia, among other chronic conditions. Data were collected using self-administered questionnaires in Arabic and English. Trained medical and public health students oversaw data collection. The full questionnaire is available in Appendix A. A multistage sampling strategy was applied: regions were first stratified by urban/rural classification, followed by purposive selection of clusters (e.g., community centers, universities), and finally, convenience sampling was used to recruit adult participants within each cluster.

### 2.4. Statistical Analysis

Data was analyzed using R software (version 4.2.3). Descriptive statistics were presented as means with standard deviations for continuous variables and frequencies with percentages for categorical variables. Bivariate analyses employed chi-square test for categorical variables and independent two-sample *t*-tests for continuous variables to compare participants with and without a DM diagnosis. To assess the independent associations between DM and potential predictors, a binary logistic regression model was fitted, with self-reported DM diagnosis (yes/no) as the dependent variable. Independent variables included sociodemographic factors, body mass index (BMI), lifestyle habits, and dietary behaviors. Continuous variables (e.g., age, BMI, sleep hours) were modeled as continuous predictors in the logistic regression, with odds ratios interpreted per unit increase. Findings are reported as odds ratios (ORs) with corresponding 95% confidence intervals (Cis) with statistical significance set at *p* < 0.05. Model performance was evaluated using Tjur’s R^2^.

### 2.5. Ethical Considerations

Ethical approval was obtained from the Standing Committee for Scientific Research at Jazan University (IRB approval No. REC-45/05/848, dated 26 November 2023). Participation was entirely voluntary and anonymous, and all participants provided written informed consent, and the study adhered to the ethical principles of the Declaration of Helsinki.

## 3. Results

### 3.1. Sociodemographic of the Study Participants

The study included 3411 Saudi participants (50.9% male, 49.1% female; mean age 34.1 ± 15.2 years) predominantly from rural areas (62%). The sample characteristics included: 54% with university education, 50% single marital status, and 25% reporting household income > 15,000 SAR/month. The mean family size was 7.0 ± 2.9 persons. Out of 3411 participants, 424 (12.4%) had diabetes. Diabetics were older (48 ± 16 vs. 32 ± 13 years), more likely male (59% vs. 50%), married (73% vs. 43%), and had larger families (7.3 ± 2.9 vs. 6.9 ± 2.9) compared to non-diabetics (all *p* < 0.05). Lower education levels were more prevalent among diabetics—illiterate (8% vs. 2%) and elementary (7% vs. 2%)—while non-diabetics more often held university degrees (55% vs. 49%). Income above 15,000 SAR/month was slightly higher in diabetics (29% vs. 27%), but non-diabetics more commonly earned 5000–10,000 SAR (19% vs. 17%; *p* < 0.05). Diabetics more often lived in owned traditional houses (28% vs. 21%), while non-diabetics preferred rented homes (13% vs. 8%; *p* < 0.05). Residence showed no significant differences (*p* > 0.05) (Table 1).

### 3.2. Anthropometric, Lifestyle, and Genetic Risk Factors

Diabetics had higher mean height (164 ± 9.7 cm vs. 163 ± 9.2 cm), weight (73 ± 16 kg vs. 66 ± 17 kg), and BMI (27 ± 5.4 vs. 25 ± 5.5; all *p* < 0.05). Physical activity levels did not differ significantly (*p* > 0.05), with 43% diabetics and 38% of non-diabetics reporting no activity, 18% of diabetics versus 22% of non-diabetics meeting recommended activity levels. Cigarette smoking was reported by 13% versus 8% and shisha use by 8% versus 5% (*p* < 0.05). Among diabetics, 86% report a family history of diabetes compared to 47% of non-diabetics (*p* < 0.001), and 8% of diabetics versus 5% of non-diabetics reporting family history of dyslipidemia (*p* < 0.05) (Table 2).

### 3.3. Dietary Behaviors

Diabetics were more likely to choose low-fat meats (60% vs. 49%), avoid sugary foods (46% vs. 36%), and select low-fat products (38% vs. 32%) (*p* < 0.05). However, consumption of whole grains and daily intake of fruits and vegetables did not differ significantly (*p* > 0.05). Both groups demonstrated poor adherence to dietary recommendations, with 74% of diabetics and 78% of non-diabetics consuming less than five daily servings of fruits and vegetables (Table 3)

### 3.4. Multivariate Predictors of Diabetes

Key independent predictors included age (OR = 1.07 per year, 95% CI:1.06–1.09), male sex (OR = 1.65, 95% CI:1.26–2.16), Family history of diabetes (OR = 7.68, 95% CI:5.67–10.57), Traditional housing (OR = 1.82 vs. rented, 95% CI:1.11–3.05). In contrast, whole grain consumption was protective (OR = 0.67, 95% CI:0.52–0.85). BMI, physical activity, and most dietary factors were non-significant in the adjusted model. The model explained variance (Tjur’s R^2^ = 0.21) (Table 4 and Figure 1).

## 4. Discussion

This study examined the interplay of nutritional behaviors, genetic predispositions, and sociodemographic factors influencing DM among Saudi adults. Both modifiable (diet, lifestyle, tobacco use) and non-modifiable (age, gender, education, family history) determinants were analyzed, underscoring DM’s multifactorial nature. The observed diabetes prevalence (12.4%) is broadly consistent with IDF estimates and national reports from Saudi settings [24,25].

### 4.1. Age and Gender as Non-Modifiable Determinants

Age was a key non-modifiable factor: participants with DM were significantly older than those without (48 ± 16 vs. 32 ± 13 years; OR = 1.07 per year, 95% CI: 1.06–1.09; *p* < 0.05). National and community data show higher diabetes prevalence at older ages and a stepwise increase across age bands [26]. Regional evidence from Kuwait and Oman likewise identifies age as a dominant risk factor in Gulf populations [27,28]. Genetic susceptibility may partly shape age-related risk; Arab-population meta-analysis reports higher frequencies of diabetogenic variants such as SLC30A8 (rs13266634) [29]. Broader national surveillance adds contextual risk patterns relevant for older adults [30], while large cohort analyses consistently demonstrate age as a strong predictor of T2DM (e.g., NHANES; visual analytics of chronic disease trends) [31,32]. Saudi community estimates further support the age gradient [33]. Preventive interventions are effective in midlife: intensive lifestyle programs reduced incident diabetes in landmark trials (DPP; Da Qing) [34,35], and a Saudi study highlighted underutilization of metformin for prevention despite its proven efficacy [36]. Methodological work from UK Biobank supports robust ascertainment approaches for diabetes phenotypes in large datasets [37].

Gender differences were also evident: males had higher diabetes prevalence (59% vs. 41%; OR = 1.65, 95% CI: 1.26–2.16; *p* < 0.05), consistent with Saudi data showing gender-based differences in dietary and lifestyle patterns [38]. Service organization can influence detection, particularly in women with PCOS-related dysglycemia; both national screening programs and Saudi research highlight gaps in early identification [39,40,41,42]. Specialized clinic models in Riyadh show promise for improving detection and earlier diagnosis [43]. Biologically, greater visceral adiposity in males and estrogenic protection in premenopausal females contribute to sex differences in metabolic risk [44,45], with local lifestyle contexts further shaping opportunities for physical activity and healthy choices [6,46].

### 4.2. Education, Marital Status, and Income: Sociocultural Influences

Lower educational attainment was associated with higher diabetes prevalence—consistent with international gradients linking socioeconomic position to metabolic risk and premature mortality [47,48,49]. In Saudi Arabia, higher education correlates with better prevention and glycemic management, likely reflecting greater health literacy, access, and adherence [48].

Marital status also mattered: prevalence was higher in married than single adults, mirroring patterns reported in Kuwait and Oman [50,51]. Within Saudi families, polygamous households show distinctive metabolic profiles, suggesting behavioral and stress-related mechanisms [52]. National survey and community-based data indicate three plausible pathways, (1) behaviors (e.g., lower physical activity, greater fast-food consumption), (2) sociocultural stressors (including financial strain), and (3) reproductive/biological factors, linked to women’s metabolic risk [53,54,55,56]. These mechanisms position marriage as a biosocial determinant of diabetes.

Household income displayed a non-linear association, with higher diabetes prevalence at both low and high ends of the income distribution and lower prevalence in middle-income groups [57]. National expenditure data show greater outlays on energy-dense foods among higher-income households and suboptimal purchase patterns for fruits and vegetables [57,58]. Policy levers may help real-world Gulf programs and price policies suggest that targeted subsidies and price controls can improve diet quality and cardiometabolic outcomes [59,60]. Integrated monitoring across expenditure and health indicators can support equitable, food environment–oriented prevention [57].

### 4.3. Environmental and Genetic Risk Contributors

Residential context also related to risk: participants with DM were more likely to live in owned traditional houses, whereas those without DM more often lived in rented apartments—patterns that may reflect generational and lifestyle differences. In Gulf settings, higher socioeconomic status is frequently associated with greater fast-food consumption and lower physical activity, and adolescent studies illustrate early adoption of Westernized dietary patterns [6,61]. Although urbanization is often linked to higher diabetes risk via lifestyle transitions, this association is not uniform across studies and settings [62].

Family history was strongly associated with diabetes in this study (OR = 7.68, 95% CI: 5.67–10.57; *p* < 0.001), aligning with Saudi and regional reports showing substantially higher odds among individuals with affected first-degree relatives [33,58]. International and genetic-epidemiology evidence supports strong familial aggregation and heritability of T2DM [18,63,64]. Saudi studies emphasize local genetic architecture and cardiometabolic clustering, including diabetogenic variants (e.g., KCNJ11, SLC30A8) in consanguineous populations [65,66]. These findings support risk-stratified approaches—earlier screening in high-risk families and exploration of pharmacogenomic tailoring as the evidence base evolves [67].

### 4.4. Lifestyle Factors: Obesity, Inactivity, Tobacco Use, and Diet

BMI was higher in participants with DM (27 vs. 25 kg/m^2^; *p* < 0.01), consistent with obesity patterns in national/international surveillance and cohort studies, and with mechanistic pathways linking excess adiposity to insulin resistance and T2DM [68,69,70]. Psychosocial stressors common in regional workplaces and obstructive sleep apnea may compound adiposity-related risk [71,72]. Sedentary behavior remains prevalent nationally and contributes substantially to NCD burden [73].

Tobacco use was also higher in DM (cigarettes 13% vs. 7%; shisha 8% vs. 4%; *p* < 0.05). Regional and international meta-analyses link smoking with higher T2DM risk, and multinational studies demonstrate elevated cardiovascular risk among smokers [9,74,75]. Biological pathways include inflammation, oxidative stress, and β-cell dysfunction, impairing insulin signaling and secretion [76]. Pragmatic models for translating intensive lifestyle programs to routine practice offer additional levels for population-level prevention [75].

Dietary behaviors indicated likely post-diagnosis adaptation (e.g., greater selection of low-fat options and avoidance of sugary foods among those with DM), yet both groups reported suboptimal fruit and vegetable intake—consistent with national data and local measurement work on dietary assessment [57,77]. These patterns reinforce the need for supportive food environments and accessible nutrition guidance.

### 4.5. Implications

This study underscores diabetes as a multifactorial disease shaped by lifestyle, genetics, and sociodemographic. Public health efforts in Saudi Arabia should prioritize early screening—particularly for individuals with a family history and lower education levels—and implement culturally tailored interventions that promote healthy diet, regular physical activity, and smoking cessation. Preventive nutrition strategies targeting high-risk groups before diagnosis are essential. Strengthening community-based programs, such as the Healthy Food Strategy and initiatives under Saudi Vision 2030, will further support these goals.

### 4.6. Strengths and Limitations

The study has several strengths, including a large sample size (n = 3411), population-based recruitment, and the integration of both modifiable and non-modifiable risk factors within a unified analytical framework. However, certain limitations should be acknowledged. The cross-sectional design limits causal inference, and the reliance on self-reported data for diabetes status, anthropometrics, and lifestyle behaviors introduces potential recall and social desirability biases. Additionally, the dietary questionnaire, although based on validated items from national surveys, did not capture quantitative nutrient intake or details of food preparation, which may affect reproducibility and the depth of dietary analysis. Furthermore, the sample was predominantly rural (62%), which may limit the generalizability of the findings to urban populations. Future longitudinal research incorporating objective clinical and genetic measurements is needed to strengthen causal understanding and enhance validity.

## 5. Conclusions

This study highlights the complex interplay of sociodemographic, genetic, lifestyle, and nutritional factors influencing diabetes risk among Saudi adults. Older age, male gender, lower education, family history, higher BMI, and tobacco use were significant determinants. Despite some positive dietary changes among diabetics, overall adherence to healthy eating and physical activity remains suboptimal. These findings emphasize the need for targeted, culturally sensitive public health strategies focusing on early screening, education, lifestyle modification, and smoking cessation to effectively address the growing diabetes burden in Saudi Arabia.

## Figures and Tables

**Figure 1 diagnostics-15-02451-f001:**
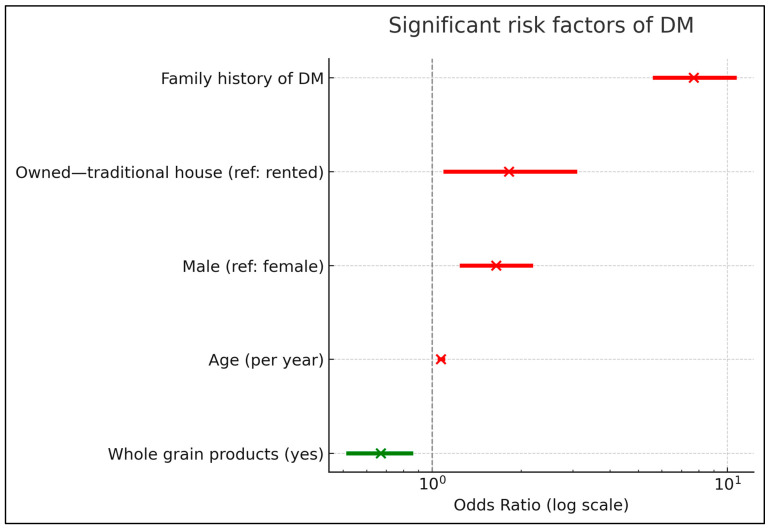
Forest plot of significant risk factors of diabetes mellitus (DM) based on multivariable logistic regression analysis. Odds ratios (ORs) with 95% confidence intervals are shown on a logarithmic scale. Red markers indicate factors associated with increased risk, while green markers indicate protective factors.

**Table 1 diagnostics-15-02451-t001:** Distribution of Sociodemographic variables by diabetes categories (n = 3411).

Variable	No		Yes		***p* Value ***
	n	Mean ± SD/Frequency (%)	n	Mean ± SD/Frequency (%)	
Age	2987	32 (13)	424	48 (16)	*p* < 0.05
Gender	2987		424		*p* < 0.05
Female	1504	50%	172	41%	
Male	1483	50%	252	59%	
Education	2987	-	424		*p* < 0.05
Elementary	66	2%	30	7%	
Illiterate	52	2%	35	8%	
Intermediate	92	3%	27	6%	
Postgraduate	106	4%	16	4%	
Secondary	1028	34%	107	25%	
University	1643	55%	209	49%	
Social Status	2987	-	424	-	*p* < 0.05
Divorced	58	2%	12	3%	
Married	1282	43%	309	73%	
Single	1601	54%	77	18%	
Widowed	46	2%	26	6%	
Income	2987		424	-	*p* < 0.05
<5000 SAR	969	32%	114	27%	
>5000 and <10,000 SAR	566	19%	73	17%	
>10,000 and <15,000 SAR	635	21%	112	26%	
>15,000 SAR	817	27%	125	29%	
Residence	2987	-	424	-	*p* > 0.05
Rural	1831	61%	268	63%	
Urban	1156	39%	156	37%	
Housing	2987	-	424		*p* < 0.05
Owned-Apartment	597	20%	83	20%	
Owned-Traditional House	619	21%	117	28%	
Owned-Villa	1375	46%	192	45%	
Rented	396	13%	32	8%	
Family Size	2987	6.9 (2.9)	424	7.3 (2.9)	*p* < 0.05

* Chi-square test for qualitative variables; independent *t*-test for quantitative variables.

**Table 2 diagnostics-15-02451-t002:** Health and lifestyle characteristics by diabetes status (n = 3411).

Variable	No		Yes		*p* Value *
	n	Mean ± SD/ Frequency (%)	n	Mean ± SD/ Frequency (%)	
Height	2987	163 (9.2)	424	164 (9.7)	*p* < 0.05
Weight	2987	66 (17)	424	73 (16)	*p* < 0.05
BMI	2987	25 (5.5)	424	27 (5.4)	*p* < 0.05
Physical activity	2987		424		*p* > 0.05
Moderate or vigorous activity for less than 30 min, 5 times per week	1210	41%	165	39%	
Moderate or vigorous activity for a minimum of 30 min, 5 times per week	653	22%	78	18%	
No physical activity is performed per week	1124	38%	181	43%	
Cigarettes	2987		424		*p* < 0.05
No	2747	92%	369	87%	
Yes	240	8%	55	13%	
Shisha	2987		424		*p* < 0.05
No	2836	95%	390	92%	
Yes	151	5%	34	8%	
Family history of DM	2987		424		*p* < 0.001
No	1582	53%	59	14%	
Yes	1405	47%	365	86%	
Family history of Dyslipidemia	2987		424		*p* < 0.05
No	2829	95%	390	92%	
Yes	158	5%	34	8%	

* Chi-square test.

**Table 3 diagnostics-15-02451-t003:** Nutrition-related behaviors stratified by diabetes status (n = 3411).

Variable	No		Yes		*p* Value *
	n	Mean ± SD/ Frequency (%)	n	Mean ± SD/ Frequency (%)	
Consumption of whole grain products	2987		424		*p* > 0.05
No	1564	52%	225	53%	
Yes	1423	48%	199	47%	
Minimum of consuming 5 servings of fruits and vegetables per day	2987		424		*p* > 0.05
No	2327	78%	313	74%	
Yes	660	22%	111	26%	
Choosing low fat meats	2987		424		*p* < 0.05
No	1516	51%	169	40%	
Yes	1471	49%	255	60%	
Avoidance of high sugar food	2987		424		*p* < 0.05
No	1924	64%	231	54%	
Yes	1063	36%	193	46%	
Choosing low fat products	2987		424		*p* < 0.05
No	2038	68%	265	62%	
Yes	949	32%	159	38%	

* Chi-square test.

**Table 4 diagnostics-15-02451-t004:** Determinants of diabetes mellitus among participants: multiple logistic regression (n = 3411).

Variable	Odds Ratios	CI	*p* Values
Age	1.07	1.06–1.09	*p* < 0.001
Gender			
Male [Ref: Female]	1.65	1.26–2.16	*p* < 0.001
Education [Ref: Illiterate]			
Elementary	1.20	0.58–2.49	*p* > 0.05
Intermediate	1.56	0.72–3.36	*p* > 0.05
Postgraduate	0.94	0.38–2.29	*p* > 0.05
Secondary	1.22	0.63–2.38	*p* > 0.05
University	0.93	0.49–1.79	*p* > 0.05
Social status [Ref: Single]			
Divorced	0.88	0.37–1.95	*p* > 0.05
Married	1.10	0.72–1.69	*p* > 0.05
Widowed	1.03	0.47–2.23	*p* > 0.05
Income [Ref: Less than 5000 SAR]			
>5000 and <10,000 SAR	0.69	0.47–1.01	*p* > 0.05
>10,000 and <15,000 SAR	0.75	0.51–1.10	*p* > 0.05
>15,000 SAR	0.98	0.67–1.42	*p* > 0.05
Residence [Ref: Rural]			
Urban	0.96	0.74–1.24	*p* > 0.05
Housing [Ref: Rented house]			
Owned-apartment	1.55	0.95–2.58	*p* > 0.05
Owned-traditional house	1.82	1.11–3.05	*p* < 0.05
Owned-villa	1.48	0.93–2.41	*p* > 0.05
Family size	0.98	0.94–1.02	*p* > 0.05
BMI	1.02	1.00–1.04	*p* > 0.05
Physical activity [Ref: No physical activity]			
Moderate or vigorous activity for a less than 30 min, 5 times per week	1.03	0.78–1.35	*p* > 0.05
Moderate or vigorous activity for a minimum of 30 min, 5 times per week	1.10	0.78–1.54	*p* > 0.05
Reference [yes]			
Smoking	1.00	0.68–1.47	*p* > 0.05
Shisha	1.08	0.67–1.70	*p* > 0.05
Sleep hours	1.05	0.98–1.13	*p* > 0.05
Consumption of whole grain products	0.67	0.52–0.85	*p* < 0.05
Minimum of consuming 5 serving of fruits and vegetables per day	1.09	0.83–1.43	*p* > 0.05
Choosing low fat meats	1.07	0.83–1.37	*p* > 0.05
Avoidance of food high in sugar	1.13	0.88–1.44	*p* > 0.05
Choosing low fat products	0.97	0.75–1.25	*p* > 0.05
Family history of DM	7.68	5.67–10.57	*p* < 0.05
Family history of Dyslipidemia	1.06	0.67–1.64	*p* > 0.05

Observations: 3411, Tjur’s R^2^ = 0.21.

## Data Availability

The data presented in this study is available on request from the corresponding author.

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
