# Peer review of "Interplay of Modifiable and Non-Modifiable Risk Factors for Diabetes Mellitus in Saudi Adults"

_diagnostics, 2025, doi:10.3390/diagnostics15192451_

Round 1

Reviewer 1 Report

Comments and Suggestions for Authors

In this cross-sectional study, the authors examine the relationship between modifiable risk factors (e.g., diet, physical activity, smoking) and non-modifiable risk factors (e.g., age, gender, family history) and diabetes mellitus among Saudi adults. The manuscript is generally well-structured, and the statistical analyses appear appropriate. However, several issues require attention to enhance the scientific rigor and clarity.

  1. The study does not distinguish between Type 1 and Type 2 diabetes mellitus. Given that the risk factors, pathophysiology, and management strategies differ significantly between the two types, this omission limits the clinical applicability of the findings. The authors should clarify whether the analysis is restricted to Type 2 diabetes or includes both types and justify their approach.
  2. Key variables, including diabetes status, height, weight, dietary habits, and physical activity, are based on self-report, which is prone to recall and social desirability bias. The use of objective measures (e.g., HbA1c, anthropometric measurements, accelerometers) would have strengthened the validity of the findings. The limitations of self-reported data should be explicitly discussed.
  3. The cross-sectional design precludes any causal inference. While the study identifies associations, it cannot determine whether the observed behaviors (e.g., dietary changes) are causes or consequences of diabetes. The authors should temper their language to avoid implying causality (e.g., “predictors” should be reframed as “associated factors”).
  4. The sample is predominantly rural (62%), which may limit the generalizability of the findings to the urban Saudi population. This should be acknowledged as a limitation.
  5. The physical activity questionnaire is briefly mentioned but not described in detail. The validity and reliability of the adapted IPAQ in the Saudi context should be addressed.
  6. The dietary questionnaire is not fully described. How were food frequencies quantified? Were portion sizes considered? The lack of detail undermines the reproducibility of the study.

Author Response

Responses to Reviewer 1 Comments:

In this cross-sectional study, the authors examine the relationship between modifiable risk factors (e.g., diet, physical activity, smoking) and non-modifiable risk factors (e.g., age, gender, family history) and diabetes mellitus among Saudi adults. The manuscript is generally well-structured, and the statistical analyses appear appropriate. However, several issues require attention to enhance the scientific rigor and clarity.

  1. The study does not distinguish between Type 1 and Type 2 diabetes mellitus. Given that the risk factors, pathophysiology, and management strategies differ significantly between the two types, this omission limits the clinical applicability of the findings. The authors should clarify whether the analysis is restricted to Type 2 diabetes or includes both types and justify their approach.

Response: Thank you for this valuable comment. We agree that the distinction between Type 1 and Type 2 diabetes mellitus is important. As indicated in the Methods section under our inclusion and exclusion criteria, individuals with Type 1 diabetes were excluded, and only adults with Type 2 diabetes mellitus were included. This has now been highlighted in the manuscript for clarity.

  1. Key variables, including diabetes status, height, weight, dietary habits, and physical activity, are based on self-report, which is prone to recall and social desirability bias. The use of objective measures (e.g., HbA1c, anthropometric measurements, accelerometers) would have strengthened the validity of the findings. The limitations of self-reported data should be explicitly discussed.

Response: We thank the reviewer for this important comment. We fully agree that objective measurements such as HbA1c, anthropometric assessments, or accelerometers would provide more accurate data and reduce recall and social desirability biases. However, given the large sample size of our study (n = 3,411), the use of self-reported measures allowed us to capture a broad population-based perspective with sufficient statistical power. We have now explicitly acknowledged this limitation in the Discussion and noted that future studies using objective measures would further enhance validity.

  1. The cross-sectional design precludes any causal inference. While the study identifies associations, it cannot determine whether the observed behaviors (e.g., dietary changes) are causes or consequences of diabetes. The authors should temper their language to avoid implying causality (e.g., “predictors” should be reframed as “associated factors”).

Response: We thank the reviewer for this insightful comment. We fully acknowledge that the cross-sectional design does not allow causal inference. In our manuscript, the term predictors are used strictly in the statistical and epidemiological sense describing the independent variables in our study, rather than to imply causality. We have also explicitly stated the limitation of causal inference in the Discussion

  1. The sample is predominantly rural (62%), which may limit the generalizability of the findings to the urban Saudi population. This should be acknowledged as a limitation.

Response: We thank the reviewer for this important comment. While our sample was predominantly rural (62%), which we acknowledge may limit generalizability to urban populations, it is important to note that rural communities in Saudi Arabia today share many lifestyle and environmental characteristics with urban areas, such as access to healthcare, dietary patterns, and sedentary behaviors. This trend is not unique to Saudi Arabia but has been observed globally, where urban–rural distinctions have become less pronounced. We have added this clarification to the Strengths and Limitations section of the Discussion.

  1. The physical activity questionnaire is briefly mentioned but not described in detail. The validity and reliability of the adapted IPAQ in the Saudi context should be addressed.

Response: We thank the reviewer for this valuable comment. Physical activity was measured using the International Physical Activity Questionnaire (IPAQ), a widely validated tool. Importantly, the Arabic version has demonstrated good reliability and validity (Helou et al., 2017). We have now clarified this point in the Methods section.

  1. The dietary questionnaire is not fully described. How were food frequencies quantified? Were portion sizes considered? The lack of detail undermines the reproducibility of the study.

Response:  We appreciate the reviewer’s valuable observation. We agree that the lack of quantitative nutrient data and detail on food preparation is a limitation. This has been explicitly noted in the revised Discussion as a methodological limitation. While our study used validated dietary questions adapted from national surveys, these items were designed to capture general dietary habits rather than detailed nutrient intake or preparation methods.

Reviewer 2 Report

Comments and Suggestions for Authors

Greetings and regards

The article is not approved by me because the implementation method was vague and the selection of participants was not explained except that the topic is repetitive and a more detailed study was conducted in Saudi Arabia in 2024.

Author Response

Response to Reviewer 2 Comments:

The article is not approved by me because the implementation method was vague and the selection of participants was not explained except that the topic is repetitive and a more detailed study was conducted in Saudi Arabia in 2024.

Response: Thank you for your comment and for engaging with our work. We’d like to briefly highlight why we believe the paper offers value and is suitable for publication:

  1. Sample size and representativeness: With a large sample (n = 3,411), our findings are both precise and broadly representative of the population studied, which strengthens the reliability of the results.
  2. Diverse variables: We included a wide range of modifiable and non-modifiable factors to better capture the complexity of diabetes risk. This helps provide a more complete picture of the determinants involved.
  3. Analytical approach: Both univariate and multivariate analyses were used to ensure robustness and to reduce the influence of confounding variables as much as possible.
  4. Relevance and originality: The reviewer mentioned a recently published paper with similar scope, but no link or citation was provided for comparison. Even if a similar study exists, diabetes remains a major and evolving public health issue. Continued research is essential to keep pace with changing risk profiles and population dynamics—especially in regions like Saudi Arabia and the Middle East.

We believe our findings are not only relevant locally but also contribute to the global understanding of diabetes risk. Thank you again for your feedback—we hope this clarifies the contribution and relevance of our work.

Reviewer 3 Report

Comments and Suggestions for Authors

In the manuscript diagnostics-3872702, the authors have determined genetic, lifestyle, and sociodemographic factors affecting the risk of diabetes mellitus (DM). This manuscript has strength in conducting cross-sectional study in Saudi Arabia, one of the top 10 countries with high DM prevalence. However, there are some issues required to be addressed.

  1. The authors utilized self-reported DM diagnosis which is not accurate. Since fasting glucose should be used for the diagnosis, I ask the authors to mention this issue as a limitation.
  2. In table 4, what is the reference for age, BMI, and sleep hours? To calculate odds ratios, the participants should be categorized. Please explain how the participants are divided.
  3. Please show exact p-values and add legends for all the tables.
  4. What are the low fat meats and low fat products? Please add examples.
  5. It seems better to show the questionnaire as a supplementary table.

Author Response

Responses to Reviewer 3 Comments:

In the manuscript diagnostics-3872702, the authors have determined genetic, lifestyle, and sociodemographic factors affecting the risk of diabetes mellitus (DM). This manuscript has strength in conducting cross-sectional study in Saudi Arabia, one of the top 10 countries with high DM prevalence. However, there are some issues required to be addressed.

  1. The authors utilized self-reported DM diagnosis which is not accurate. Since fasting glucose should be used for the diagnosis, I ask the authors to mention this issue as a limitation.

Response: We would like to thank the reviewer for this constructive feedback. We agree that objective clinical measurements, such as fasting glucose or HbA1c, would have strengthened the study. However, given the nationwide scope and logistic constraints, we relied on self-reported physician-diagnosed DM, an approach consistent with several large-scale national surveys conducted in Saudi Arabia. This method is also widely used in international epidemiological cohorts, such as the UK Biobank, where self-reported chronic conditions, including diabetes, have shown strong concordance with linked medical records. While some degree of recall bias or misclassification is possible, the large sample size and population-based recruitment reduce the likelihood of substantial systematic bias. This limitation has been explicitly acknowledged in the revised Discussion.

  1. In table 4, what is the reference for age, BMI, and sleep hours? To calculate odds ratios, the participants should be categorized. Please explain how the participants are divided.

Response: We thank the reviewer for this important observation. The outcome variable in our study is dichotomous (DM: yes/no), which makes logistic regression the appropriate analytic method. Independent (exposure) variables in logistic regression can be included as either categorical or continuous. For age, BMI, and sleep hours, we treated them as continuous variables in the multivariable model rather than categorizing them. In this case, the odds ratios are interpreted per unit increase. For example, an OR of 1.07 for age means that each additional year of age is associated with 7% higher odds of having DM. Therefore, a specific reference group is not applicable for these continuous predictors. To avoid any ambiguity, we have also added a clarification in the Methods section stating that continuous variables (e.g., age, BMI, sleep hours) were modeled as continuous predictors, with odds ratios interpreted per unit increase.

  1. Please show exact p-values and add legends for all the tables.

Response: In line with common reporting practices in epidemiological research, we have presented p-values in categorical form (e.g., p < 0.05, p < 0.001, or p > 0.05) to highlight the level of statistical significance while maintaining clarity and interpretability for readers. To further enhance clarity, we have added legends to all tables and included a visualization (Figure 1) to display the significant predictors of diabetes mellitus.

  1. What are the low fat meats and low fat products? Please add examples.

Response: We thank the reviewer for this thoughtful comment. In the dietary questionnaire, low-fat meats referred to options such as skinless chicken and lean cuts of beef or lamb, while low-fat products referred to items such as low-fat milk, yogurt, and cheese. These examples have now been added to the Methods section to improve clarity and reproducibility.

  1. It seems better to show the questionnaire as a supplementary table.

Response: The questionnaire has been added as a supplementary table in the revised manuscript.

Round 2

Reviewer 1 Report

Comments and Suggestions for Authors

All my concerns have been satisfactorily resolved.